# Emotional stability is related to 2D:4D and social desirability in women: Possible implications on subjective well-being and psychopathology

Ángel Rodríguez-Ramos[1,2,3,4]*, Juan Antonio Moriana[2,3,4], Francisco García-Torres[2,3,4], Manuel Ruiz-Rubio[1,3,4]

1 Department of Genetics, University of Córdoba, Córdoba, Spain, 2 Department of Psychology, University of Córdoba, Córdoba, Spain, 3 Maimónides Biomedical Research Institute of Córdoba (IMIBIC), Córdoba, Spain, 4 University Hospital Reina Sofía of Córdoba, Córdoba, Spain

* angel.rguezramos@gmail.com

**Data Availability Statement:** All relevant data are within the manuscript and its Supporting Information files.

## Abstract

Emotional stability-Neuroticism is a complex construct influenced by genetics and environmental factors. Women tend to exhibit higher neuroticism scores than men, which may be associated with an increased risk of suffering from some common mental conditions. Some authors have pointed out the influence of sex hormones, since they induce sexual differentiation of the brain that can lead to sex-specific behaviors. 2D:4D digit ratio is commonly used as a marker of prenatal sex hormones. In this study we analyzed whether there was an association between 2D:4D and personality measured through the BFQ in a homogeneous sample of 101 young women college students. We found a positive association between 2D:4D and emotional stability, as well as with its subdimensions emotion control and impulse control. This association could be quadratic and nonlinear. However, no association was found with the other four dimensions. We also measured anxiety, depression and global life satisfaction, variables related to neuroticism. We observed that emotional stability is positively associated to social desirability and global life satisfaction, and negatively related to anxiety and depression. On the other hand, we did not find any association between 2D:4D and anxiety, depression, and global life satisfaction. These results can be linked to other aspects such as subjective well-being and psychopathological symptoms. This study may help to better understand how these constructs are related and could lead to future projects to elucidated how these variables influence personality.

## Introduction

Personality and psychopathology have been related since the time of the ancient Greeks and today the existence of this relationship has been evidenced by different studies [1–3]. Over the years different models have been postulated to describe and explain personality, among which the most recognized is the Five-Factor Model. This model was developed empirically through

**Funding:** The research work was supported by the group BIO-272, grant to Manuel Ruiz-Rubio from Junta de Andalucía, Spain (https://www. juntadeandalucia.es); group GC20 'Genetics and Behavioral Diseases' from the IMIBIC, Spain (https://www.imibic.org); and by the University of Córdoba, Spain (https://www.uco.es).

**Competing interests:** The authors have declared that no competing interests exist.

lexical studies of the trait terms in different languages, converging in a five-factor structure consisting of openness, conscientiousness, extraversion, agreeableness and neuroticism [4]. Using this model sex differences have been found in some traits, with agreeableness and neuroticism being the most highlighted dimensions [5, 6]. Neuroticism is a stable personality trait included in almost all major models of personality and has been shown to predict many mental and physical disorders [7], and is one of the most important predictors of subjective well-being [8]. Neuroticism is moderately heritable (~ 47%) [7]. People with high neuroticism levels experience more negative emotions, tend to worry and have less control over their impulses and desires [7]. Previous studies have shown that women tend to be more neurotic than men [5, 6] and these differences become smaller with age, although small differences are present even in old age [9].

The fact that women tend to exhibit higher neuroticism scores could moderate the association between being a woman and an increased risk of suffering from some common mental disorders, such as mood and anxiety disorders [10, 11]. The causes of these differences are not yet fully understood, but some authors point to the influence of sex hormones [12]. These act on the nervous system through activational and organizational effects. Activational effects are transient and facilitate behavior in specific contexts depending on current hormone levels, generally during adolescence and adulthood. In contrast, organizational effects are permanent and lead to the final organization and structure of the nervous system [13]. These occur in specific stages of neural development during sensitive periods where the hormone effects are maximal, inducing sexual differentiation of the brain. There are two main periods during which sex hormones modify the brain and influence human psychological sex differences, early prenatal development and puberty [14, 15]. The mechanisms involved in this differentiation cause a dimorphic organization of certain brain regions that results in sex-specific behaviors [16]. In women there are also other periods with high fluctuations in hormone levels that seem to modify the brain, affecting certain aspects of behavior. This is specially frequent during pregnancy and menopausal transitions; but even short-term fluctuations could as occur in the menstrual cycle, although these changes often revert to baseline [17, 18].

The second to fourth digit ratio (2D:4D) is commonly used as a marker of the balance between prenatal testosterone and estrogen [19]. 2D:4D describes the relative length of the index finger to the ring finger which it is constant throughout life and tends to be lower in men than in women [20]. Several studies have estimated that 2D:4D is highly heritability (50–80%) [21]. In order to explain the effect of prenatal sex hormones on the digit ratio, multiple studies have attempted to find an association between 2D:4D and the CAG-repeat length of the androgen receptor gene, in which the longest alleles are related to a decreased androgen receptor transcription and sensitivity to testosterone [22]. These studies have shown several discrepancies. However, what seems to be clear is that there is not a significant relationship [23] or if there is, it is a weak association with this polymorphism [24]. So, although several studies have found an association between 2D:4D and direct measurements of prenatal sex hormones in human [25], this mechanism is still unknown. Additionally, the relationship between 2D:4D and testosterone levels in adults is not significant [23, 24]. Due to these discrepancies, some authors are skeptical about whether 2D:4D is a good candidate as a marker of early hormonal environment and instead propose other measures [26, 27].

Despite this, so many studies have found correlations between 2D:4D and multiple variables, including psychological aspects such as personality. Regarding to personality research, the Five-Factor Model have been the most widely investigated, using different tests. With the NEO Five-Factor Inventory, one study found a positive association with neuroticism, a negative association with agreeableness and a close tendency association with extraversion, all of them only with the right 2D:4D of women [28]. Concerning neuroticism, it has been observed

a significant positive association of 2D:4D of both hand and only the left hand with neuroticism in women from two different populations, German and Chinese respectively [29]. A relationship with openness across the entire sample with both 2D:4D and separately between the right 2D:4D in men has also been described [30]. A research including four ethnic groups found a positive association between right 2D:4D and agreeableness regardless of the population origin [31]. Using the Five Factor Personality Inventory, a significant association was described in a whole sample and separately in both sexes between right 2D:4D and agreeableness, and also with conscientiousness but only in the whole sample [32]. With the Big Five Inventory, an investigation found a significant weak association of 2D:4D with extraversion and openness to experiences, and a tendency with agreeableness but not with other traits [33]. A recent study with data from a large population, using three items of the Big Five Inventory-SOEP, found a relationship between neuroticism and the mean of both 2D:4D and separately with the left hand. However, they have proposed that this association could be quadratic and nonlinear [34]. An internet study across 23 nations found a positive correlation between both 2D:4D and the cultural dimensions of uncertainty avoidance in both sexes and aggregate neuroticism only in men [35]. Two Italian investigations with the Big Five Questionnaire (BFQ) found a positive relationship with conscientiousness and agreeableness in a sample of skydivers [36], and a positive relationship with conscientiousness in a sample of cavers [37]. Recently, using the ten-item personality inventory a positive relationship have been found between the right 2D:4D and openness to new experiences and a negative one with disorganization and carelessness [38]. Additionally, in the same study they found a negative relationship between the left 2D:4D and dependable self-disciplined. Furthermore, in addition to using the Five-Factor Model, some studies have investigated the relationship with 2D:4D using other personality models such as the Eysenck Personality Model [39], the Cattell's Sixteen Personality Factor Model [40, 41], and the HEXACO Personality Model [42].

Therefore, despite there are several studies that have shown an association between 2D:4D and personality, contradictory results have been found. Among these studies, only two have used the BFQ [36, 37], both with heterogeneous samples of people who choose high-risk activities, with less than 50 people and only measuring the 2D:4D ratio in the right hand. Consequently, these results could be biased. For this reason, we decided to study whether there was an association between 2D:4D and personality using the BFQ where we expect to find a small but significant association with some of the five personality traits, especially in neuroticism that had been one of the most reported. With this objective, we chose a homogeneous group of young women university students and we measured 2D:4D in both hands. Only women were included in this study because they generally tend to show higher scores on neuroticism, which could be interesting for predicting mental health. Although we studied the five dimensions of the Five-Factor Model, we have focused mainly on neuroticism and we have also considered some inter-related constructs such as anxiety, depression and global life satisfaction, since they are important for different mental disorders that have previously been related to 2D:4D [43–46].

## Materials and methods

### Participants

Healthy young women university students from the Faculty of Sciences and the Faculty of Education of the University of Córdoba (Spain), were randomly selected to participate in the present research. In order to homogenize the sample, we established an age range between 18 and 28 years old and some participants were rejected due to older age or missing data. Finally, one hundred and one participants were selected. The characteristics of the sample are shown

**Table 1. Characteristics of the sample.**

| Variable | N | % | Mean | SD |
|---|---|---|---|---|
| *Faculty of* | | | | |
| Sciences | 51 | 50.5 | | |
| Education | 50 | 49.5 | | |
| *Sex* | | | | |
| Women | 101 | 100.0 | | |
| *Age* | 101 | | 21.02 | 2.05 |
| *Sexual orientation* | | | | |
| Heterosexual | 93 | 92.1 | | |
| Homosexual | 2 | 2.0 | | |
| Bisexual | 6 | 5.9 | | |
| *2D:4D* | | | | |
| Left-Hand | 101 | | 1.009 | .038 |
| Right-Hand | 101 | | .985 | .042 |

in Table 1. They provided written informed consent for an anonymous research. The study was approved by the ethical research committee of the Reina Sofía University Hospital (Córdoba, Spain).

## Procedures and measures

The present study was carried out during practical classes where the students responded to different self-report tests, including BFQ, STAI, BDI and SWLS. In addition, a photograph of both hands was taken to measure the 2D:4D.

**Big Five Questionnaire (BFQ).** The BFQ is a personality test based on the Five-Factor Model, that identifies five fundamental dimensions in the human personality. The Spanish version of the BFQ [47] consists in 132 multiple-choice items of a 5-points Likert scale. The scores are used to measure 5 different dimensions: Energy, Friendliness, Conscientiousness, Emotional Stability and Openness, which refer to the factors generally labeled Extraversion, Agreeableness, Conscientiousness, Neuroticism and Openness to experience, respectively [48]. Each dimension is organized into two facets which are: Energy (Dynamism and Dominance), Friendliness (Cooperativeness/Empathy and Politeness), Conscientiousness (Scrupulousness and Perseverance), Emotional Stability (Emotion Control and Impulse Control) and Openness (Openness to Culture and Openness to Experiences). The BFQ also includes a Lie scale that measures social desirability with the purpose of identifying people who tend to offer a distorted profile to project an erroneous image of themselves in a more or less intentional way. These dimensions and their subdimensions are measured on a continuum where an individual might be anywhere between the two extremes. Raw scores were transformed into standardized T-scores (mean 50, SD 10) to run the analysis. Reference values, internal consistency and scores of self-report questionnaires have been compiled in Table 2.

**State-Trait Anxiety Inventory (STAI).** The STAI is a psychological questionnaire designed to assess the level of state and trait anxiety [52]. State anxiety reflects a transient emotional state at a specific moment and that changes over time; while trait anxiety is a relatively stable personality characteristic over time and different situations which refers to a tendency to experience anxiety. The Spanish version of the STAI has a good internal consistency (.80 for state anxiety and .88 for trait anxiety) [49]. It consists in 40 items, 20 to assess state anxiety and 20 for trait anxiety. In each item, participants responded with a 4-point Likert scale from 0 to 3

**Table 2. Internal consistency, reference values and scores from self-report questionnaires.**

| Variable | α* | Raw Scores (mean) | Raw Scores (SD) | Reference raw values* | T Mean | T SD |
|---|---|---|---|---|---|---|
| *Personality (BFQ)* | | | | | | |
| *Energy* | .75 | 76.98 | 9.02 | 75.02 | 52.22 | 9.371 |
| Dynamism | .68 | 41.09 | 6.08 | 40.03 | 51.49 | 9.506 |
| Dominance | .66 | 35.89 | 4.70 | 34.99 | 51.75 | 8.141 |
| *Friendliness* | .73 | 86.40 | 10.64 | 83.45 | 53.49 | 10.994 |
| Cooperativeness/Empathy | .60 | 45.29 | 5.75 | 43.61 | 53.59 | 11.393 |
| Politeness | .62 | 41.11 | 6.17 | 39.84 | 52.66 | 10.783 |
| *Conscientiousness* | .79 | 90.06 | 9.97 | 80.49 | 59.28 | 9.512 |
| Scrupulousness | .71 | 43.27 | 6.74 | 38.23 | 57.52 | 9.956 |
| Perseverance | .76 | 46.79 | 5.55 | 42.26 | 57.15 | 8.798 |
| *Emotional Stability* | .87 | 67.21 | 13.56 | 64.96 | 51.62 | 10.159 |
| Emotion control | .79 | 34.18 | 7.08 | 33.14 | 51.30 | 9.266 |
| Impulse control | .78 | 33.03 | 7.79 | 31.82 | 51.60 | 10.871 |
| *Openness* | .76 | 87.24 | 10.42 | 83.58 | 53.73 | 10.445 |
| Openness to Culture | .67 | 43.09 | 6.63 | 41.54 | 52.57 | 10.629 |
| Openness to Experiences | .64 | 44.15 | 5.79 | 42.05 | 53.68 | 10.087 |
| *Lie* | .77 | 28.67 | 4.64 | 28.82 | 50.09 | 7.354 |
| *Anxiety (STAI)* | | | | | | |
| State | .80 | 14.75 | 9.29 | 15.34 | | |
| Trait | .88 | 21.28 | 9.72 | 20.15 | | |
| *Depression (BDI)* | .87 | 6.17 | 5.77 | 9.4 | | |
| *Global Life Satisfaction (SWLS)* | .88 | 26.11 | 5.05 | 24.23 | | |

* Taken from: BFQ [47]; STAI [49]; BDI [50] and SWLS [51].

points as "very much", "likely", "not so much," or "not at all", indicating in which extent they agree or disagree with the statements. Higher total scores indicate greater anxiety.

**Beck's Depression Inventory (BDI).** The BDI is an inventory created to measure characteristic attitudes and symptoms of depression [53]. The Spanish version of the BDI-II ($\alpha = .88$) [50] consists of 21 multiple-choice questions with four different self-report statements rated on a scale from 0 to 3. The total score was calculated by adding the score of all the responses, with higher scores indicating greater symptom severity. The total score ranges from 0 to 63 points and different cutoff ranges have been established: 0–13 minimal depression, 14–19 mild depression, 20–28 moderate depression, and 29–63 severe depression.

**Satisfaction With Life Scale (SWLS).** The SWLS is a psychological scale used to measure global life satisfaction, a cognitive component of the subjective well-being [54]. Like the original, the Spanish version of the SWLS ($\alpha = .88$) [51] consists of five statements in which the participants refer to their lives using a 7-point Likert scale from 1 "completely disagree" to 7 "completely agree". Scores are added together to obtain an overall score where high scores indicate higher levels of life satisfaction and vice versa. Different categories have been established according to the scores obtained: 31–35 extremely satisfied; 26–30 satisfied; 21–25 slightly satisfied; 20 neutral; 15–19 slightly dissatisfied; 10–14 dissatisfied; and 5–9 extremely dissatisfied.

**2D:4D digit ratio measurement.** A photograph of both hands was taken with a digital camera [55]. The hands were held in supination with the full extension of the fingers. 2D:4D was computed as the quotient of the index and the ring fingers. The lengths of these fingers were measured from the bottom crease to the tip of the finger using ImageJ software. In case

there were more than one crease, the most proximal one was used [28]. Measurements were carried out by two independent raters and the mean of both was used. Inter-rater reliability was calculated using an intraclass correlation coefficient (ICC) in which were assumed that people effects are random and measurements effects are fixed. The ICC obtained for the average of the two raters were 0.89 for the left hand and 0.90 for the right, so they presented an excellent reliability [56].

## Statistical analysis

Statistical analysis was performed using IBM SPSS Statistics 25. Bivariate correlations between the different quantitative variables analyzed were performed using the Pearson's correlation coefficient. A multiple linear regression was carried out through the stepwise method to stablish the explanatory power of the different variables related to emotional stability on it as the dependent variable. The results were accepted as significant at $p \leq .05$.

## Results

We found a positive correlation between the left 2D:4D and the dimension emotional stability (Table 3), as well as with its two subdimensions, emotion control and impulse control. On the contrary, no association was found with the other four dimensions: energy, friendliness, conscientiousness and openness, nor any of its subdimensions; and neither with anxiety, depression, global life satisfaction or social desirability. On the other hand, no significant relationships were found with the right 2D:4D. Similar results have been observed with the square of the 2D:4D of both hands (Table 3). Although a significant correlation could be obtained randomly given the large number of correlations performed, we believe that it is very unlikely that significant correlations, if they were random, would occur exactly in the expected cases (emotional stability and its subdimensions).

Referring to emotional stability, some associations have been observed with social desirability, anxiety, depression and global life satisfaction, all of them showing a high significant association (Table 4). All these relationships were also found for its two subdimensions. Likewise, social desirability, in addition to the association with emotional stability, showed a significant positive association with global life satisfaction and negative with anxiety and depression (Fig 1).

Furthermore, we evaluated the explanatory power of these variables on emotional stability. The results showed that the significant variables explained 54.7% of the variance in emotional stability ($R^2 = .547$, F(3,101) = 41.240, $p < .001$). The significant predictor variables were trait anxiety, social desirability, and the square of left 2D:4D. The contribution of each one to the model has been collected in Table 5.

## Discussion

### 2D:4D is positively related to emotional stability but not with the other four dimensions

In the present study we described for the first time a positive correlation between the left 2D:4D and emotional stability, and with its subdimensions emotion control and impulse control. These correlations were not found with the right 2D:4D. Therefore, our results indicate that lower 2D:4D is associated with lower emotional stability (higher neuroticism) in women, and consequently we presume that is also associated with higher prenatal testosterone exposure. These results are partially in disagreement with previous studies which found a relationship between 2D:4D and the neuroticism-emotional stability dimension but seem to be in the

**Table 3. Statistical results of correlation analysis between 2D:4D and self-report variables.**

| Self-report variable | Left 2D:4D | | Right 2D:4D | | Left 2D:4D$^2$ | | Right 2D:4D$^2$ | |
|---|---|---|---|---|---|---|---|---|
| | r(99) | p | r(99) | p | r(99) | p | r(99) | p |
| Energy | -.070 | .484 | -.126 | .211 | -.071 | .483 | -.124 | .217 |
| Dynamism | -.003 | .978 | -.048 | .635 | -.004 | .969 | -.046 | .648 |
| Dominance | -.132 | .189 | -.181 | .070 | -.131 | .190 | -.181 | .070 |
| Friendliness | .032 | .748 | -.011 | .912 | .027 | .790 | -.015 | .878 |
| Cooperativeness/Empathy | -.076 | .448 | -.065 | .519 | -.082 | .413 | -.073 | .469 |
| Politeness | .082 | .412 | .021 | .832 | .079 | .434 | .022 | .826 |
| Conscientiousness | -.058 | .567 | -.027 | .785 | -.058 | .567 | -.026 | .797 |
| Scrupulousness | -.062 | .539 | -.024 | .811 | -.062 | .535 | -.020 | .844 |
| Perseverance | -.037 | .715 | -.032 | .753 | -.036 | .719 | -.034 | .735 |
| Emotional stability | .271 | .006** | .108 | .284 | .271 | .006** | .110 | .272 |
| Emotion control | .251 | .012* | .132 | .190 | .250 | .012* | .136 | .176 |
| Impulse control | .236 | .017* | .050 | .616 | .237 | .017* | .051 | .609 |
| Openness | -.044 | .663 | .025 | .808 | -.048 | .634 | .019 | .848 |
| Openness to Culture | .016 | .877 | .082 | .415 | .013 | .894 | .079 | .434 |
| Openness to Experiences | -.070 | .484 | -.052 | .604 | -.076 | .450 | -.058 | .567 |
| Social Desirability | .162 | .106 | .134 | .182 | .163 | .103 | .135 | .179 |
| State Anxiety | .005 | .963 | .091 | .364 | .013 | .897 | .094 | .349 |
| Trait Anxiety | -.157 | .117 | -.079 | .431 | -.153 | .127 | -.080 | .425 |
| Depression | -.068 | .500 | -.075 | .458 | -.065 | .520 | -.080 | .427 |
| Global life satisfaction | .072 | .472 | -.037 | .716 | .071 | .482 | -.034 | .734 |

* $p \leq .05$

** $p \leq .01$.

opposite direction [28, 29, 35, 39–42]; or, conversely, have found no relationship between these variables [30–33, 36–38]. In most of the previous studies neuroticism was measured through the Neo Five-Factor Inventory, and here we measured emotional stability through the BFQ. Although the dimensions neuroticism in the Neo Five-Factor Inventory and emotional stability in the BFQ have shown a high negative correlation between them and, therefore, both seem to measure the same dimension [48]; their division into several subdimensions are different, probably grouping different traits in different ways in each test. In the regression model, the square of the left 2D:4D has been shown to explain a part of the variance in emotional

**Table 4. Statistical results of correlation analysis between emotional stability, and its subdimensions, with the other self-report variables.**

| Reported variable | Emotional Stability | | Impulse control | | Emotion control | |
|---|---|---|---|---|---|---|
| | r(99) | p | r(99) | p | r(99) | p |
| Social Desirability | .477 | .000*** | .480 | .000*** | .380 | .000*** |
| State Anxiety | -.474 | .000*** | -.348 | .000*** | -.503 | 000*** |
| Trait Anxiety | -.676 | .000*** | -.530 | .000*** | -.700 | .000*** |
| Depression | -.529 | .000*** | -.393 | .000*** | -.571 | .000*** |
| Global life satisfaction | .456 | .000*** | .370 | .000*** | .464 | .000*** |

* $p \leq .05$

** $p \leq .01$

*** $p \leq .001$.

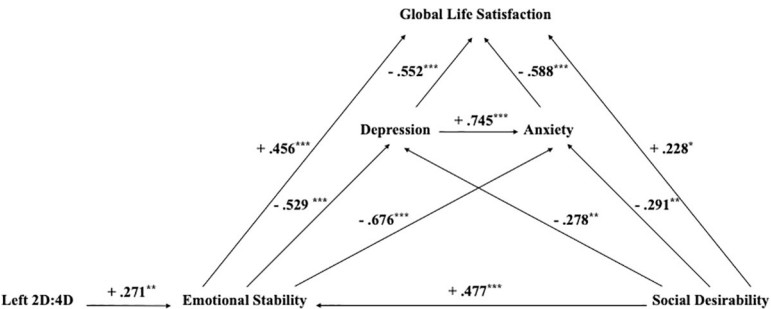

**Fig 1. Main associations found related to emotional stability as personality trait.** Arrows indicate that there is a significant correlation between the two linked constructs and the value indicates the Pearson's correlation coefficient between them. ** $p \leq .01$; *** $p \leq .001$.

stability, as recently has been proposed [34]. So, it would be interesting to continue investigating this possible nonlinear relationship in subsequent studies. Therefore, taking into account all the data, it is the first time that a positive relationship has been shown between 2D:4D and emotional stability, which may contribute to a better understanding of the relationship between 2D:4D and emotional stability, and some psychopathological conditions as mentioned below.

On the other hand, although we did not find any significant association between 2D:4D and the other four dimensions and these results agree with several previous investigations, other studies have shown controversial results. For example, referring to agreeableness, some studies, like ours, have not found a significant association with 2D:4D [29, 30, 37, 42], others have found a positive correlation [31, 32, 36] and others a negative correlation [28, 33]. This inconsistency is another example that more research is needed in this field. These discrepancies could be due to the different tests used to measure personality and also the high heterogeneity between the samples used. So, as our results support, in the future would be important to use homogeneous samples and evaluate possible nonlinear relationships, as well as optimize the best type of study to understand how exactly 2D:4D is related to personality. In any case, what seems clear is that if there is a relationship between personality and 2D:4D, it possibly explains only a small percentage of the individual differences observed in the different traits.

## The association between 2D:4D and emotion control and impulse control could explain its relationship with mental disorders

Neuroticism has previously been associated with anxiety, depression, and lower subjective well-being [7, 8]. Subjective well-being is established through three related components: frequent positive affect, infrequent negative affect and cognitive evaluations such as global life satisfaction [57]. It has been proposed that the influence of personality on subjective well-being accounts between 39% and 63% [58], with emotional stability-neuroticism being one of the most important predictors of both cognitive and affective components of subjective well-being [8]. In line with these previous observations, we found that emotional stability is positively

**Table 5. Regression model of emotional stability.**

| Predictor variable | *B* | *SE B* | β | *t* | *p* |
|---|---|---|---|---|---|
| Trait anxiety | -.597 | .074 | -.571 | -8.068 | < .001 |
| Social Desirability | .398 | .098 | .288 | 4.058 | < .001 |
| Left 2D:4D$^2$ | 18.131 | 9.087 | .137 | 1.995 | .049 |

related to global life satisfaction, and negatively with depression and anxiety. In turn, global life satisfaction correlates negatively with anxiety and depression. Thus, when people exhibit higher levels of emotional stability, they will experience greater global life satisfaction and less anxiety and depression, which would result in greater subjective well-being [8].

Anxiety, depression and subjective well-being are important variables for suicide risk and several mental disorders, including anxiety-, mood- and eating disorders, all of which have been reported to be related to 2D:4D [43–46]. Despite these observations, we did not find any association between these variables and 2D:4D, which agrees with the limited literature on this topic. With regard to anxiety, most associations with 2D:4D in the general population have been found in men but not in women [44, 59]. In relation to depression, some discrepancies have been observed, with some studies showing associations between depression and 2D:4D only in men [60, 61], only in women [45] or no association [34, 39]. Regarding subjective well-being, most of the studies have not found an association with 2D:4D; however, a recent article have pointed out to a possible inverse U-shaped relationship [62]. Although more research is needed in this field, we suggest that 2D:4D, instead of acting directly on anxiety, depression, and subjective well-being, would act on emotional stability, specifically on impulse control and emotion control, that indirectly would influence these variables. Thus, prenatal testosterone would affect emotion control and impulse control which would moderate the association between 2D:4D and some types of mental disorders.

On the other hand, we have also found that emotional stability is positively correlated with social desirability. At the same time social desirability is negatively related to anxiety and depression and positively with global life satisfaction. However, we did not find a significant association between social desirability and 2D:4D, which coincide with previous reports that studied this association [36, 37, 63, 64]. Social desirability is an additional predictor of the components of subjective well-being, along with personality traits [8]. It is described as the tendency of an individual to offer a biased answer to personality items in order to present himself in a more socially desirable way than his real answer would be [65]. Otherwise, other authors have indicated that more than a scale to measure response bias, social desirability measures a positive and prosocial personality trait [66], where self-control might be implicated in socially desirable behaviors and it is linked to the universal need to belong [67]. Our results are in agreement with previous studies that have investigated personality using the BFQ and had found a positive association between emotional stability and social desirability, measured through the lie-scale [36, 37]. Moreover, they found a negative correlation between social desirability and anxiety, and with the tendency to suffer from mood disorders. Likewise, other studies have shown a negative association between social desirability and neuroticism, and also with alexithymia, which seems to be moderated by higher neuroticism scores [68]. Therefore, greater social desirability will lead to better control of impulses and emotions in order to develop socially desirable behaviors to do the right thing, which is often a socially desirable act. In this sense, low scores on social desirability will be related to some psychopathological symptoms such as hostility, depression or social withdrawal [68]. On the other hand, people with high scores that do not show high levels of self-control, and also high self-monitors characterized by chameleon-like and in somehow manipulative behaviors, will only seek the approval of others, without thinking about the correct behavior [67].

## 2D:4D and impulse control: Possible implications in psychopathology

Impulse control is a subdimension of emotional stability that refers to the ability to control discontent, irritation and anger [48]. This construct is closely linked to others, such as impulsivity, which is also a main symptom of ADHD, aggression or any disruptive behavior. Although

we did not directly measure these constructs, other studies have investigated them and have found some results that might be related to ours.

Some studies have investigated the association between 2D:4D and ADHD, which is characterized by inattention, impulsivity, and in some cases, hyperactivity. These symptoms are the same for children and adults. However, some studies suggest that in adults, rather than inattention, it is more specific to consider executive dysfunction [69], and support that emotional dysregulation is a hallmark symptom of ADHD in adults [70]. Similarly, ADHD has presented a certain personality profile generally characterized by high scores in neuroticism and low scores in conscientiousness, and also has shown lower extraversion and agreeableness than healthy controls [71, 72]. So, the various symptoms are associated with personality traits in a yet unknown interaction. Previous work has associated prenatal exposure to testosterone with increased hyperactivity-impulsivity in girls and not in preschool-age boys. They proposed that hyperactivity-impulsivity might be sensitive to the organizational effects of sex hormones and, thus, high prenatal exposure to testosterone may increase risk of ADHD in girls [73].

Impulsivity appears to be higher during childhood and adolescence than adulthood [74], and an early manifestation of hyperactivity-impulsivity could predispose to disruptive behavior problems [75]. General disruptive behaviors such as aggression, hyperactivity and conduct problems have been linked to prenatal exposure to higher levels of testosterone, and to hyperactivity and conduct problems in preschool girls [73]. In the same direction, other research has shown that a low digit ratio is associated with anger, hostility, physical aggression (and verbal aggression approaching significance), total sensation seeking, thrill and adventure seeking [76], less precautionary behavior [36, 37], and with risk taking in women [77]. This last observation is supported with data showing that in women, optimism might moderate the association between right 2D:4D and financial risk taking [78]. Therefore, although more studies should be carried out, we can speculate that higher levels of prenatal testosterone in women could lead to a lower emotional stability, and also a lower impulse control, which could increase the risk of ADHD, aggression and other disruptive behaviors. These personality traits are likely to appear during childhood and remain even into adulthood.

A recent study has shown that *AR* rs6152 and *OPRM1* rs2075572, SNPs of the androgen receptor and the β-endorphin receptor genes respectively, might affect impulsivity mediated by 2D:4D digit ratio [79]. Moreover, they found a relationship between *AR* rs6152 and left hand 2D:4D only in women. In fact, *AR* rs6152 has been related to polycystic ovary syndrome [80], which in turn have been related to higher levels of neuroticism [81]. Furthermore, some research have found an association between this polymorphism and the level of expression of the androgen receptor in both men and women [82]. Therefore, although previous studies with the *AR* CAG-repeat length and 2D:4D have shown that there is no significant relationship [23], and that there does not appear to be a relationship between levels of sex hormone in adults with 2D:4D [24], testosterone seems to regulate impulsivity in some way, at least in women. One hypothesis could be that testosterone may have a greater effect in women, since they exhibit more variable levels of prenatal testosterone than men, who generally exhibit higher levels of prenatal testosterone [73]. Hence, it would be interesting to further study all these variables together in order to better understand these relationships.

In summary, emotional stability-neuroticism is a personality dimension that have been linked to suffering from anxiety and various mood disorders. Previous studies have shown that women tend to be more neurotic than men, which could moderate the increased risk of them from suffering from some common mental disorders. Higher levels of neuroticism have also been shown to negatively affect subjective well-being. Emotional stability-neuroticism is influenced by several aspects, including genetics, 2D:4D digit ratio and social aspects such as social desirability. 2D:4D is considered an indirect measure of prenatal testosterone, which could

modify the brain when it is still developing. Prenatal testosterone is likely to have a greater effect in women as they exhibit more variable levels of prenatal testosterone than men, especially in aspects such as impulse control and emotion control. This influence on the subdimensions of emotional stability might be the basis for other relationships previously found between 2D:4D and some mental disorders. On the other hand, with respect to social aspects, people with higher levels of social desirability may have better self-control to act in a socially desirable way.

## Limitations

Some limitations of this study should be mentioned. First, we have used 2D:4D as somatic marker of prenatal testosterone, while its molecular mechanism and the exact relationship between them are not yet fully understood. On the other hand, given the activational effects of sex hormones, these may influence personality, therefore future studies should monitor the day of the menstrual cycle, the use of oral contraceptives and whether they are or have been pregnant, especially recently. Another limitation of our study is that the sample only includes university students and may not reflect the general population. Finally, although a power analysis was performed and most of the results agree with previous studies, we cannot discard completely the type I errors from multiple testing. Nevertheless, despite these limitations, this study could help to better understand how emotional stability and its subdimensions are influenced by prenatal testosterone and social aspects such as social desirability and could lead to future projects to elucidate the mechanisms involved in this relationship.

## Conclusions

Emotional stability is a complex construct that is influenced by genetics and environmental factors. We have found results that suggest that in women 2D:4D could be positively related to emotional stability and to its subdimensions, impulse control and emotion control. In fact, these associations could be quadratic and nonlinear. Given that 2D:4D has been proposed as an indirect measure of prenatal testosterone, these results can be interpreted as that this hormone may affect emotional stability and its subdimensions, at least in women. The effect of prenatal testosterone on personality may, in turn, influence other aspects, including psychopathological symptoms such as impulsivity in ADHD and aggression. Furthermore, we have found that social desirability might also affect emotional stability and its subdimensions. Thus, people with higher levels of social desirability would better control their impulses to act in a socially desirable way, linked to the universal need to belong.

## Supporting information

**S1 Data.**
(XLSX)

## Author Contributions

**Conceptualization:** Ángel Rodríguez-Ramos, Juan Antonio Moriana, Manuel Ruiz-Rubio.

**Data curation:** Ángel Rodríguez-Ramos.

**Formal analysis:** Ángel Rodríguez-Ramos, Juan Antonio Moriana, Francisco García-Torres.

**Funding acquisition:** Juan Antonio Moriana, Manuel Ruiz-Rubio.

**Investigation:** Ángel Rodríguez-Ramos, Juan Antonio Moriana, Francisco García-Torres, Manuel Ruiz-Rubio.

**Methodology:** Ángel Rodríguez-Ramos, Juan Antonio Moriana, Manuel Ruiz-Rubio.

**Project administration:** Manuel Ruiz-Rubio.

**Supervision:** Juan Antonio Moriana, Manuel Ruiz-Rubio.

**Validation:** Ángel Rodríguez-Ramos, Juan Antonio Moriana, Francisco García-Torres, Manuel Ruiz-Rubio.

**Writing – original draft:** Ángel Rodríguez-Ramos, Manuel Ruiz-Rubio.

**Writing – review & editing:** Ángel Rodríguez-Ramos, Juan Antonio Moriana, Francisco García-Torres, Manuel Ruiz-Rubio.

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
