## [Decision Letter · Decision Letter 0]

18 Jan 2021

PONE-D-20-29650

Emotional stability is related to 2D:4D and social desirability in women: Possible implications on subjective well-being and psychopathology

PLOS ONE

Dear Dr. Rodríguez-Ramos,

Thank you for submitting your manuscript to PLOS ONE. After careful consideration, we feel that it has merit but does not fully meet PLOS ONE’s publication criteria as it currently stands. Therefore, we invite you to submit a revised version of the manuscript that addresses the points raised during the review process.

Apart from Reviewers#1’s comments (especially those related to line 278, i.e. how results of this study could contribute to the better understanding of that relationship), I would like to add the following:

A justification for the sample size should be given (optimally – as a consequence of the power analysis).Like Reviewer#1 I have a problem with the rationale for introducing anxiety, depression, and subjective well-being. Once the rationale is clearly formulated the way how these variables should be analyzed will be clear. If the idea is to investigate how a broad definition of the domain would correlate with 2D:4D, then one or two latent factors should be extracted from these variables (emotional control, impulse control, depression, anxiety, social desirability) and correlate with 2D:4D. If the idea is to see which aspect of ES is crucial in understanding 2D:4D, then these variables should be partialled out from the BFQ ES and this residualized BFQ ES related to 2D:4D. It could be something else, but - to my understanding - there is no much sense of relating BFQ  ES to these variables independently from its relationships with 2D:4D because 1) we know that they correlate and 2) it is not the goal of your study.

We look forward to receiving your revised manuscript.

Kind regards,

Goran Knezevic

Academic Editor

PLOS ONE

Journal Requirements:

2. For studies involving humans categorized by age, disease/disabilities, religion, sexual orientation, or other socially constructed groupings, authors should: 1) Explicitly describe their methods of categorizing human populations, 2) Define categories in as much detail as the study protocol allows, 3) Justify their choices of definitions and categories, 4) Explain whether (and if so, how) they controlled for confounding variables such as socioeconomic status, nutrition, environmental exposures, or similar factors in their analysis, and 5) Update outmoded terms and potentially stigmatizing labels to more current, acceptable terminology. Examples: “suffering” should be changed to “diagnosed with” or “having xxx”.

3. For more information on PLOS ONE's expectations for statistical reporting, please see https://journals.plos.org/plosone/s/submission-guidelines.#loc-statistical-reporting. Please update your Methods and Results sections accordingly.

Reviewers' comments:

Reviewer's Responses to Questions

**Comments to the Author**

1. Is the manuscript technically sound, and do the data support the conclusions?

Reviewer #1: Yes

Reviewer #2: Yes

2. Has the statistical analysis been performed appropriately and rigorously? 

Reviewer #1: Yes

Reviewer #2: Yes

3. Have the authors made all data underlying the findings in their manuscript fully available?

Reviewer #1: Yes

Reviewer #2: Yes

4. Is the manuscript presented in an intelligible fashion and written in standard English?

Reviewer #1: Yes

Reviewer #2: Yes

5. Review Comments to the Author

Reviewer #1: I find the article acceptable with some needed changes and revisions. Also, I would suggest a proofreading of the text in grammatical and stylistic sense.

Additional comments to the Author(s)

Minor revisions:

38: satisfaction, variables instead of satisfaction; variables

38-39: stability is positively related instead of stability could be positively related

134: carelessness instead of careless

245: comma or colon after dimensions

To be explained:

155: In this section, I recommend to describe briefly the reasons for the selection of the instruments other than BFQ (Anxiety, Depression and Satisfaction with Life).

165: why religion?

250: explain why the significant correlation can't be obtained by random given a rather large number of all correlations. For example, I believe that it is very unlikely that significant correlations, if they were random, would occur exactly in the expected cases (emotional stability).

272: significant predictors instead of significant

278: In this section (Discussion), it would make sense to discuss the lack of 2D:4D correlation with variables that are substantially associated with the emotional stabililty – neuroticism (anxiety, depression, life satisfaction and social desirability).

Also, I miss a more thorough clarification of controversial results in studies investigating the relationship between 2D:4D and emotional stability. In what extent, the results of this study could contribute to the better understanding of that relationship?

310: variance; explain which variance

433: to instead of with

434: whose link...; this part of the sentence is not clear. It should be reformulated.

434: In fact...; make the sentence more precise.

437-440: this personality trait... ; the last sentence is clumsy, it would be better to split entire formulation in several simple sentences.

Reviewer #2: Thank you for the opportunity to review the manuscript entitled: Emotional stability is related to2D:4D and social desirability in 1women: Possible implications on subjective well-being and psychopathology.

1. The structure of the manuscript was carefully prepared and organized. The manuscript is methodologically sound and the data generated supported the conclusions.

2. In the present study, the statistical analysis has been performed appropriately and rigorously , and the results and conclusions are consistent with the main objective.

3. All relevant data were included in the manuscript.

4. The manuscript is written in sound English

The article fulfills all of the requirements listed above and is ready for publication.

6. PLOS authors have the option to publish the peer review history of their article (what does this mean?). If published, this will include your full peer review and any attached files.

Reviewer #1: No

Reviewer #2: No

---

## [Author Response · Author response to Decision Letter 0]

3 Feb 2021

RESPONSE TO REVIEWERS

Apart from Reviewers#1’s comments (especially those related to line 278, i.e. how results of this study could contribute to the better understanding of that relationship), I would like to add the following:

1. A justification for the sample size should be given (optimally – as a consequence of the power analysis).

In order to estimate the necessary sample size in our study, we have considered the correlation between neuroticism and right 2D:4D in females shown by Fink et al. (2004) (r = .292), with a power of 80% and alpha = .05 (two tails). We selected a priori analysis using the G*Power program, applying an exact test (correlation: bivariate normal model). The results suggest that we would need 72 participants. So, theoretically our sample (n = 101) can be considered big enough to detect this correlation.

We have included in Limitations a sentence regarding to this request (Page 17, Lines 463-465): “Finally, although a power analysis was performed and most of the results agree with previous studies, we cannot discard completely the type I errors from multiple testing”.

Reference: 

Fink B, Manning JT, Neave N. Second to fourth digit ratio and the ‘big five’ personality factors. Pers Individ Dif. 2004;37: 495–503. doi: 10.1016/J.PAID.2003.09.018.

2. Like Reviewer#1 I have a problem with the rationale for introducing anxiety, depression, and subjective well-being. Once the rationale is clearly formulated the way how these variables should be analyzed will be clear. If the idea is to investigate how a broad definition of the domain would correlate with 2D:4D, then one or two latent factors should be extracted from these variables (emotional control, impulse control, depression, anxiety, social desirability) and correlate with 2D:4D. If the idea is to see which aspect of ES is crucial in understanding 2D:4D, then these variables should be partialled out from the BFQ ES and this residualized BFQ ES related to 2D:4D. It could be something else, but - to my understanding - there is no much sense of relating BFQ ES to these variables independently from its relationships with 2D:4D because 1) we know that they correlate and 2) it is not the goal of your study.

In addition to measuring personality and social desirability using the BFQ, we decided to measure anxiety, depression, and subjective well-being because these variables are involved in many pathological conditions such as anxiety-, mood- and eating disorders. Interestingly, these disorders have previously been related to 2D:4D. That is why we decided to measure them trying to investigate in an integrative way together with emotional stability, which is also involved in them. Nevertheless, we agree that this was not strictly stated in the manuscript as it was, but we believe that in the new version it has been highlighted. To do this, we have modified the order of the sections of the Discussion and we also have adapted one of these sections to dedicate it to this topic “The association between 2D:4D and emotion control and impulse control could explain its relationship with mental disorders” (Page 13, Line 332). 

Reviewer #1: I find the article acceptable with some needed changes and revisions. Also, I would suggest a proofreading of the text in grammatical and stylistic sense.

We have carefully reviewed the text and made some changes that we think have improved the manuscript. 

Additional comments to the Author(s)

Minor revisions:

Line 38: “satisfaction, variables” instead of “satisfaction; variables”.

Corrected

Line 38-39: “stability is positively related” instead of “stability could be positively related”.

Corrected

Line 134: “carelessness” instead of “careless”.

Corrected. Now on Page 5, Line 135.

Line 245: comma or colon after dimensions. 

Corrected. Now on Page 10, Line 256.

To be explained:

155: In this section, I recommend to describe briefly the reasons for the selection of the instruments other than BFQ (Anxiety, Depression and Satisfaction with Life).

In addition to the fact that these variables have been highly related with emotional stability-neuroticism dimension, we have measured anxiety, depression and global life satisfaction because these are important for different mental disorders that have been previously related to 2D:4D. We have referred this in the manuscript (Page 5, Lines 152-156):

“Although we studied the five dimensions of the Five-Factor Model, we have focused mainly on neuroticism and we have also considered some inter-related constructs such as anxiety, depression and global life satisfaction, since they are important for different mental disorders that have previously been related to 2D:4D (43-46)”.

References:

43. Quinton SJ, Smith AR, Joiner T. The 2nd to 4th digit ratio (2D:4D) and eating disorder diagnosis in women. Pers Individ Dif. 2011;51: 402–405. doi: 10.1016/j.paid.2010.07.024.

44. Evardone M, Alexander GM. Anxiety, Sex-Linked Behaviors, and Digit Ratios (2D:4D). Arch Sex Behav. 2009;38: 442–455. doi:10.1007/s10508-007-9260-6.

45. Smedley KD, McKain KJ, McKain DN. 2D:4D digit ratio predicts depression severity for females but not for males. Pers Individ Dif. 2014;70: 136–139. doi: https://doi.org/10.1016/j.paid.2014.06.039.

46. Lenz B, Kornhuber J. Cross-national gender variations of digit ratio (2D:4D) correlate with life expectancy, suicide rate, and other causes of death. J Neural Transm. 2018;125: 239–246. doi: 10.1007/s00702-017-1815-7.

Line 165: why religion?

We collected religion as a sociodemographic variable; however, we agree that this variable is not relevant in this study. So, we have removed it from the Table 1 (Page 6, Line 169).

Line 250: explain why the significant correlation can't be obtained by random given a rather large number of all correlations. For example, I believe that it is very unlikely that significant correlations, if they were random, would occur exactly in the expected cases (emotional stability).

Following your suggestion, we have included this sentence in the Results section (Page 10, Line 260-263):

“Although a significant correlation could be obtained randomly given the large number of correlations performed, we believe that it is very unlikely that significant correlations, if they were random, would occur exactly in the expected cases (emotional stability and its subdimensions)”.

Line 272: “significant predictors” instead of “significant”.

Corrected. Now on Page 11, Line 286.

Line 278: In this section (Discussion), it would make sense to discuss the lack of 2D:4D correlation with variables that are substantially associated with the emotional stabililty – neuroticism (anxiety, depression, life satisfaction and social desirability).

We have included in Results a sentence referring to the associations between 2D:4D and anxiety, depression and global life satisfaction (Page 10, Lines 256-258):

“On the contrary, no association was found with the other four dimensions: energy, friendliness, conscientiousness and openness, nor any of its subdimensions; and neither with anxiety, depression, global life satisfaction or social desirability”.

We have also modified the last section of the Discussion, including the title, to discuss these findings together with what we have described in the previous version: “The association between 2D:4D and emotion control and impulse control could explain its relationship with mental disorders” (Page 13, Line 332).

New references:

59. De Bruin EI, Verheij F, Wiegman T, Ferdinand RF. Differences in finger length ratio between males with autism, pervasive developmental disorder-not otherwise specified, ADHD, and anxiety disorders. Dev Med Child Neurol. 2006;48: 962–965. doi:10.1017/S0012162206002118.

60. Martin SM, Manning JT, Dowrick CF. Fluctuating Asymmetry, Relative Digit Length, and Depression in Men. Evol Hum Behav. 1999;20: 203–214. doi: https://doi.org/10.1016/S1090-5138(99)00006-9.

61. Bailey AA, Hurd PL. Depression in men is associated with more feminine finger length ratios. Pers Individ Dif. 2005;39: 829–836. doi: https://doi.org/10.1016/j.paid.2004.12.017.

62. Nye JVC, Bryukhanov M, Polyachenko S. 2D:4D and individual satisfaction: Evidence from the Russian social survey. Pers Individ Dif. 2019;142: 85–89. doi: https://doi.org/10.1016/j.paid.2019.01.031.

63. Schwarz S, Mustafić M, Hassebrauck M, Jörg J. Short- and long-term relationship orientation and 2D:4D finger-length ratio. Arch Sex Behav. 2011;40: 565–574. doi: 10.1007/s10508-010-9698-9.

64. Schwerdtfeger A, Heims R, Heer J. Digit ratio (2D:4D) is associated with traffic violations for male frequent car drivers. Accid Anal Prev. 2010;42: 269–274. doi: 10.1016/j.aap.2009.08.001.

Also, I miss a more thorough clarification of controversial results in studies investigating the relationship between 2D:4D and emotional stability. In what extent, the results of this study could contribute to the better understanding of that relationship?

We described for the first time that a positive relationship has been described between 2D:4D and emotional stability, as we now mention in the manuscript Page 12, Line 296. Moreover, this relationship has been found with its subdimensions emotion control and impulse control. The arguments about how these results could contribute to the better understanding of that relationship have been included throughout the Discussion and are summarized on Page 16, Lines 448-452: 

“Prenatal testosterone is likely to have a greater effect in women as they exhibit more variable levels of prenatal testosterone than men, especially in aspects such as impulse control and emotion control. This influence on the subdimensions of emotional stability might be the basis for other relationships previously found between 2D:4D and some mental disorders”, 

Line 310: variance; explain which variance

We have substituted the word ‘variance’ for ‘individual differences observed in the different traits’ (Page 13, Lines 328-330):

“In any case, what seems clear is that if there is a relationship between personality and 2D:4D, it possibly explains only a small percentage of the individual differences observed in the different traits”.

Line 433: “to” instead of “with”

Corrected, now on Page 17, Line 473.

Line 434: whose link...; this part of the sentence is not clear. It should be reformulated.

We have rewritten this sentence, now on Page 17, Lines 474-477: 

“Given that 2D:4D has been proposed as an indirect measure of prenatal testosterone, these results can be interpreted as that this hormone may affect emotional stability and its subdimensions, at least in women”. 

Line 434: In fact...; make the sentence more precise.

We have modified the sentence and changed the location in the paragraph. Now we consider that it is better understood (Page 17, Lines 472-474):

“We have found results that suggest that in women 2D:4D could be positively related to emotional stability and to its subdimensions, impulse control and emotion control. In fact, these associations could be quadratic and nonlinear”.

Line 437-440: this personality trait... ; the last sentence is clumsy, it would be better to split entire formulation in several simple sentences.

We have modified this sentence and divided it into two easier ones (Page 17, Lines 479-482):

“Furthermore, we have found that social desirability might also affect emotional stability and its subdimensions. Thus, people with higher levels of social desirability would better control their impulses to act in a socially desirable way, linked to the universal need to belong”.

---

## [Editor Report · Decision Letter 1]

25 Feb 2021

Emotional stability is related to 2D:4D and social desirability in women: Possible implications on subjective well-being and psychopathology

PONE-D-20-29650R1

Dear Dr. Rodríguez-Ramos,

We’re pleased to inform you that your manuscript has been judged scientifically suitable for publication and will be formally accepted for publication once it meets all outstanding technical requirements.

Kind regards,

Goran Knezevic

Academic Editor

PLOS ONE
---

## [Editor Report · Acceptance letter]

1 Mar 2021

PONE-D-20-29650R1 

Emotional stability is related to 2D:4D and social desirability in women: Possible implications on subjective well-being and psychopathology 

Dear Dr. Rodríguez-Ramos:

I'm pleased to inform you that your manuscript has been deemed suitable for publication in PLOS ONE. Congratulations! Your manuscript is now with our production department. 

Kind regards, 

on behalf of

Dr. Goran Knezevic 

Academic Editor

PLOS ONE